# Study of the Phytochemical Composition, Antioxidant Properties, and In Vitro Anti-Diabetic Efficacy of *Gracilaria bursa-pastoris* Extracts

**DOI:** 10.3390/md21070372

**Published:** 2023-06-24

**Authors:** Safae Ouahabi, El Hassania Loukili, Nour Elhouda Daoudi, Mohamed Chebaibi, Mohamed Ramdani, Ilyesse Rahhou, Mohamed Bnouham, Marie-Laure Fauconnier, Belkheir Hammouti, Larbi Rhazi, Alicia Ayerdi Gotor, Flore Dépeint, Mohammed Ramdani

**Affiliations:** 1Laboratory of Applied and Environmental Chemistry (LCAE), Faculty of Sciences, Mohammed First University, B.P. 717, Oujda 60000, Morocco; ouahabi.safae@ump.ac.ma (S.O.); e.loukili@ump.ac.ma (E.H.L.); hammoutib@gmail.com (B.H.); moharamdani2000@yahoo.fr (M.R.); 2Laboratory of Bioresources, Biotechnology, Ethnopharmacology and Health, Faculty of Sciences, Mohammed First University, B.P. 717, Oujda 60000, Morocco; nourelhoudada95@gmail.com (N.E.D.); mbnouham@ump.ac.ma (M.B.); 3Biomedical and Translational Research Laboratory, Faculty of Medicine and Pharmacy of the Fez, University of Sidi Mohamed Ben Abdellah, Fez 30000, Morocco; mohamed.chebaibi@yahoo.fr; 4Biochemistry and Biotechnology Laboratory, Faculty of Sciences, Mohamed First University, B.P. 717, Oujda 60000, Morocco; ramdanimed@yahoo.fr; 5Higher Institute of Nursing Professions and Health Techniques (ISPITSO), Oujda 63303, Morocco; ilyesse@hotmail.com; 6Laboratory of Chemistry of Natural Molecules, University of Liège, Gembloux Agro-Bio Tech. 2, Passage des Déportés, B-5030 Gembloux, Belgium; marie-laure.fauconnier@uliege.be; 7CREHEIO Centre de Recherche de l’Ecole des Hautes Etudes d’Ingénierie, Oujda 60000, Morocco; 8Université Euro-Méditerranéenne de Fès, Fez BP 51, Morocco; 9Institut Polytechnique UniLaSalle, Université d’Artois, ULR 7519, UniLaSalle, 19 Rue Pierre Waguet, BP 30313, 60026 Beauvais, France; flore.depeint@unilasalle.fr; 10Institut Polytechnique UniLaSalle, AGHYLE, UP 2018.C101, UniLaSalle, 19 Rue Pierre Waguet, BP 30313, 60026 Beauvais, France; alicia.ayerdi-gotor@unilasalle.fr

**Keywords:** *Gracilaria bursa-pastoris*, phenolic compounds, antioxidant activity, anti-diabetic properties, enzyme inhibition

## Abstract

In this study, a comparison was made of the chemical makeup of different extracts obtained from *Gracilaria bursa-pastoris*, a type of red seaweed that was gathered from the Nador lagoon situated in the northern part of Morocco. Additionally, their anti-diabetic and antioxidant properties were investigated. The application of GC-MS technology to analyze the fatty acid content of the samples revealed that linoleic acid and eicosenoic acid were the most abundant unsaturated fatty acids across all samples, with palmitic acid and oleic acid following in frequency. The HPLC analysis indicated that ascorbic and kojic acids were the most prevalent phenolic compounds, while apigenin was the most common flavonoid molecule. The aqueous extract exhibited significant levels of polyphenols and flavonoids, registering values of 381.31 ± 0.33 mg GAE/g and 201.80 ± 0.21 mg QE/g, respectively. Furthermore, this particular extract demonstrated a remarkable ability to scavenge DPPH radicals, as evidenced by its IC_50_ value of 0.17 ± 0.67 mg/mL. In addition, the methanolic extract was found to possess antioxidant properties, as evidenced by its ability to prevent β-carotene discoloration, with an IC_50_ ranging from 0.062 ± 0.02 mg/mL to 0.070 ± 0.06 mg/mL. In vitro study showed that all extracts significantly inhibited the enzymatic activity of α-amylase and α-glucosidase. Finally, molecular docking models were applied to assess the interaction between the primary phytochemicals identified in *G. bursa-pastoris* extracts and the human pancreatic α-amylase and α-glucosidase enzymes. The findings suggest that these extracts contain bioactive substances capable of reducing enzyme activity more effectively than the commercially available drug acarbose.

## 1. Introduction

As the demand for high-quality food products continues to rise among consumers, the use of advanced technology and ingredients in the development of functional foods has become a common response [1]. In accordance with the FFC (Functional Food Center), foods that offer established clinical benefits for the prevention, management, or treatment of chronic diseases are known as functional foods, and can either be natural or processed. These foods contain biologically active compounds in specific amounts that are well-defined [2]. Seaweeds, also known as marine macroalgae, have gained significant attention from researchers seeking novel bioactive ingredients for both medication and food since the year 2000 [3,4]. These organisms are divided into three categories, which are determined by analyzing their morphology, composition, and other characteristics. These groups are the chlorophytes, also known as green algae, phaeophytes, or brown algae, and rhodophytes, which are commonly referred to as red algae [5]. In traditional Japanese cuisine, seaweeds are frequently used as vegetables, in sauces, and as sushi wrappers [6] and provide a multitude of advantages to various sectors, such as food, pharmaceuticals, animal nutrition, skincare, and agrochemical industries. Pharmaceutical and food sectors, specifically, rely on seaweeds as a source of thickening and emulsifying agents (phycocolloids) [6,7,8,9,10]. Algae possess a wealth of bioactive components, including polysaccharides, minerals, vitamins, proteins, lipids, polyphenols (phenolic acid, flavonoids), pigments (fucoxanthins, astaxanthins, carotenoids), alkaloids, and terpenoids, which play a significant ecological and economic role and have numerous applications across various fields [11,12,13]. These bioactive substances exhibit various biological activities such as antimicrobial [13,14,15,16,17,18,19,20], antiviral, antifungal, insecticidal, cytotoxic, phytotoxic, antiproliferative [17,21,22], and hypolipidemic [23] properties. With a global count of between 20,000 and 30,000 species, seaweeds make up approximately 18% of the plant kingdom [24]. The chemical composition of these organisms varies depending on genus, habitat, maturity, and environmental conditions. Algae lack leaves, stems, and roots and consist primarily of a vegetative apparatus known as the thallus. They are multicellular, macroscopic marine organisms that can reproduce sexually and/or asexually. Among the diverse types of algae, 6600 species of red algae have been described [25]. They are referred to as Rhodophyceae and are found in saltwater environments deeper than 6 m. These red algae require blue light (and some green light) and contain chlorophyll a and d and pigments such as phycoerythrin and phycocyanin. They are mostly multicellular and attach to rocks or shells of mollusks. *Gracilaria* species are among the red algae and are widely distributed and commonly harvested or farmed for agar production [26,27]. *G. bursa-pastoris*, a member of the *Gracilariaceae* family, is primarily recognized as a food source rather than an herb. It is a type of edible seaweed widely consumed in various cultures around the world, particularly in Asian cuisines, and can be found in Japanese, Korean, and Chinese dishes. This species has a red or discolored appearance and a fleshy, cartilaginous texture. Its thicker cylindrical axes have a diameter of between 0.5 and 5 mm and are slightly compressed at the branching point. Branches are irregular and can reach 35 cm in length. The base of the branches remains thick, and they end in spikes. *G. bursa-pastoris* is commonly found in temperate and tropical regions of the northeast Atlantic, as well as in the Mediterranean.

Diabetes mellitus is a prevalent health condition characterized by an imbalance in sugar metabolism, leading to chronic hyperglycemia [28]. Current epidemiological studies indicate that diabetes affects 463 million individuals worldwide, with projections suggesting a rise to 700 million by 2045. This condition is generally classified into two main types: Type 1 diabetes and Type 2 diabetes, which are the most common forms globally. Over time, diabetes mellitus results in metabolic disturbances that contribute to the development of long-term complications, primarily affecting the vascular system. These complications can be broadly categorized as microvascular complications, also known as microangiopathy, and macrovascular complications, commonly referred to as macroangiopathy [29]. The onset and progression of these complications are closely linked to oxidative stress, which triggers the production of free radicals and activates various metabolic pathways [30]. In diabetic patients, the primary goal of treatment is to maintain blood glucose levels close to physiological norms to prevent and minimize the onset of diabetic complications. Treatment for Type 1 diabetes involves insulin injections, while the management of Type 2 diabetes includes lifestyle modifications, dietary changes, and oral anti-diabetic medications to enhance insulin sensitivity [31]. Despite the beneficial effects of these treatments in controlling blood sugar levels and maintaining body homeostasis, insulin therapy and anti-diabetic drugs can have side effects. The specific side effects and their severity may vary depending on the type of anti-diabetic drug and the individual’s response to it. Common side effects include gastrointestinal disorders, diarrhea, nausea, the risk of lactic acidosis, edema, heart failure, weight gain, flatulence, severe hypoglycemia, and vomiting [31,32,33,34]. Consequently, the search for natural anti-diabetic products has become a necessity [35]. Seaweeds have gained attention in the field of diabetes research. Several studies have highlighted the positive effects of seaweeds on various aspects of diabetes, including blood glucose control, insulin sensitivity, and the prevention of diabetic complications.

The study of the benthic algae along the Moroccan coast dates back to the end of the 19th century, with the work of P. K. Schousboe on the Tangier coast and subsequent revisions by E. Bornet (1892). Further studies emerged early in the 20th century, including the investigations of P. Hariot (1909–1919) on the Moroccan coast. Over the past few decades, the Moroccan coast has drawn renewed attention from researchers and authors [36]. Currently, in collaboration with academics, relevant institutions are working towards creating comprehensive studies of the algal flora along the national coastlines and preserving this valuable resource. To the best of our knowledge, the composition of *G. bursa-pastoris* seaweeds in the Nador lagoon has not been extensively studied.

This innovative research endeavors to conduct a comprehensive chromatographic examination of the fatty acids and polyphenols present in *G. bursa-pastoris* extracts through the use of advanced analytical techniques, including GC-MS and HPLC-DAD. Moreover, the study aims to assess the antioxidant capability of the extracts and evaluate their impact on pancreatic α-amylase and α-glucosidase activities. This is a landmark undertaking as it represents the first time that the composition of *G. bursa-pastoris* extracts has been characterized through GC/MS and HPLC, and the effect on pancreatic digestive enzymes has been studied. 

## 2. Results

### 2.1. Yields, Phenols and Flavonoids Contents

The phenols, flavonoids, and extraction yield of *G. bursa-pastoris* extracts were studied, and the results are presented in Table 1. The methanolic extract had the highest polyphenolic yield with 3.42 ± 0.06%, followed by the ethyl acetate extract with 1.80 ± 0.08%, both using Soxhlet extraction. The hexane extract had the lowest yield, at 1.16 ± 0.32%.

In terms of maceration extraction, the aqueous extract had the highest yield with 1.25 ± 0.06%, followed by the methanolic extract with 0.81 ± 0.26%. The yields of the ethyl acetate and hexane extracts were 0.21 ± 0.04% and 0.11 ± 0.09%, respectively. 

The total phenolic content of the extracts was determined based on its equivalent to gallic acid (GAE), as seen in Table 1. The highest total polyphenolic content was found in the aqueous extract using the maceration, with a value of 381.31 ± 0.33 mg GAE/g. This was followed by the ethyl acetate extract, with a value of 165.42 ± 0.21 mg GAE/g or 150.24 ± 0.11 mg GAE/g, using the maceration or Soxhlet method, respectively. These results suggest that both water and ethyl acetate are effective solvents for extracting phenolic compounds in *G. bursa-pastoris* using either maceration or Soxhlet methods. The methanolic extracts showed the lowest values of 37.69 ± 1.02 mg GAE/g and 28.68 ± 0.07 mg GAE/g.

The flavonoid content of the various extracts of *G. bursa-pastoris* was quantified as the quercetin equivalent (QE), as presented in Table 1. The concentrations of flavonoids were found to range from 13.70 ± 0.03 mg QE/g to 201.80 ± 0.21 mg QE/g. The highest concentration was observed in the aqueous extract and was obtained via the maceration method, followed by the ethyl acetate extract with 84.53 ± 0.07 mg EQ/g and 75.47 ± 0.02 mg EQ/g, using the maceration and Soxhlet methods, respectively.

### 2.2. Fatty Acid Analysis

The results of the GC-MS analysis presented in Table 2 showed that both the Soxhlet and maceration extraction techniques of *G. bursa-pastoris* resulted in a significant concentration of saturated fatty acids (SFAs) and unsaturated fatty acids (UFAs). Compared to terrestrial plants, seaweeds have a unique fatty acid (FA) profile.

The composition of fatty acids in the samples is expressed in terms of total fatty acids. The percentages of total saturated fatty acids (SFAs), monounsaturated fatty acids (MUFAs), and polyunsaturated fatty acids (PUFAs) varied in function of the solvent and the extraction method used, ranging from 13.01% to 59.3%.

In the hexane extract, eicosenoic acid (C20:1) was the most abundant fatty acid obtained via maceration extraction, with 53.06%, followed by palmitic acid (16:0), with also a high amount at 40.72%, and by linoleic acid, with relatively the same concentration in both extraction methods at 10.08%. However, the highest concentration of palmitic acid (50.55%) was found in the ethyl acetate extract obtained through maceration, followed by linoleic acid (C18:2), with a relatively high concentration (23.85%) obtained through Soxhlet extraction. Margaric acid (C17:0) was detected in only one extract, the ethyl acetate extract obtained through Soxhlet extraction, at a value of 16.19%. In this extract, oleic acid (C18:1) was more dominant than in the hexane extract, with concentrations of 13.01% and 10.48% in Soxhlet and maceration extraction, respectively. Stearic acid (C18:0) accounted for 4.77% to 6.43% of the total fatty acids. Figure 1 shows the distribution of SFA, MUFA and PUFA in each type of solvent and extraction process.

In addition, the hexane extract obtained through the maceration method revealed the highest unsaturated ratio, followed by the hexane extract under Soxhlet extraction. These results indicate that the hexane extract was rich in unsaturated fatty acids, notably eicosenoic acid. *G. bursa-pastoris* possesses great nutritional value in terms of human consumption.

### 2.3. HPLC Analysis of G. bursa-pastoris Extracts

High-Performance Liquid Chromatography combined with a Diode Array Detector (HPLC-DAD) was implemented to examine the chemical composition of the ethyl acetate and methanol extracts of *G. bursa-pastoris*. The results were compared with those of the standards based on the retention time and ultraviolet spectra and are displayed in Table 3. The study identified nine phenolic acids (4-hydroxy benzoic acid, chlorogenic acid, ferulic acid, cinnamic acid, gallic acid, caffeic acid, syringic acid, p-coumaric acid, and salicylic acid) and ten flavonoids (catechin, quercetin-3-glucoside, 7,3′,4′-flavon-3-ol, rutin, quercetin, luteolin, apigenin, kaempferol, flavone, and flavanone). Caffeic acid was the most abundant in both ME (M) (35.64%) and ME (S) (24.24%), while sinapic acid was the most prevalent in EAcE (M) (9.53%). EAcE (S) had a predominant amount of flavonoid, specifically 7,3′,4′-flavon-3-ol (20.68%) (Figure 2).

In all four extracts, several other compounds were discovered, including naringin chlorogenic acid, and kaempferol. Naringin was present in moderate levels, with values ranging from 8% to 21%, and chlorogenic acid was found at varying percentages, from 7% to 11% in ME and at low levels in EAcE, while kaempferol was detected in moderate amounts in each extract. Additionally, the extracts contained moderate quantities of 4-hydroxy benzoic acid, quercetin-3-glucoside, salicylic acid, quercetin, cinnamic acid, luteolin, and apigenin. While gallic acid, catechin, syringic acid, vanillin, p-coumaric acid, flavone, and flavanone were present in the extracts in low quantities. The presence of certain compounds in maceration extracts and their absence in Soxhlet extracts (and vice versa) can be attributed to the extraction conditions, including solvent, temperature, time, and the stability of natural products under certain conditions.

### 2.4. Antioxidant Activity

Antioxidants are the focus of research due to their role in preserving food quality and their potential to treat diseases associated with oxidative stress. In this study, the DPPH free radical scavenging assay and the β-carotene bleaching assay were used. The IC_50_ values for *G. bursa-pastoris* EAcE, ME, and AQE extracts are presented in Table 4. The results indicate that the best antioxidant activity was observed in the case of Soxhlet-extracted ME as well as for maceration-extracted ME, with IC_50_ values of 0.06 and 0.07 mg/mL, respectively, in comparison with the reference antioxidant BHA, which presented an IC_50_ value of 0.02 mg/mL. Additionally, AQE constituted a modest antioxidant with an IC_50_ value of 0.21 mg/mL; however, the EAC extracts revealed the lowest antioxidant activity. Caffeic acid is among the hydroxycinnamic acids used to enhance the stability of dietary products [37,38]. It constituted the most abundant compound in both maceration and Soxhlet-extracted methanol extracts (ME). The presence of this compound at a high concentration in these extracts (35.64% and 24.24%) could explain the high antioxidant potential observed in ME extracts.

### 2.5. In Vitro α-Amylase Inhibition

Figure 3 illustrates the effect of *G. bursa-pastoris* extracts on α-amylase inhibitory activity in comparison to acarbose as a positive control. In vitro studies were conducted to examine the impact of various doses of the extracts on α-amylase enzymatic activity. The results showed that all extracts significantly inhibited the α-amylase enzymatic activity at all tested doses. Among the different samples, the concentration of 2.27 mg/mL had the most active effect, exhibiting inhibitory activities of 51.83% for EAcE (M), 67.64% for EAcE (S), 78.54% for ME (M), 73.94% for ME (S), and 72.08% for AQE. IC_50_ values for each sample were determined (Table 5), revealing that the ME (M), AQE, and ME (S) extracts exhibited similar inhibitory effects (*p* < 0.05) and higher inhibitory activities than the other extracts. EAcE (S) (*p* < 0.05) and EAcE (M) (*p* < 0.05) followed. In addition, the results of the in vitro study demonstrated that all extracts significantly reduced the pancreatic α-glucosidase activity compared to acarbose (Figure 4). Among the extracts, EM (M) had the most significant effect (98.32%), followed by AQE (M) (96.68%), ME (S) (94.32%), EAcE (M) (88.32%), and EAcE (S) (83.32%) at a concentration of 2.27 mg/mL. The observed inhibitory effects may be attributed to the presence of active chemicals in the extracts that act as inhibitors of the enzymes responsible for hydrolyzing polysaccharide and disaccharide chains. These substances may be unique to α-amylase and α-glucosidase.

Upon comparing the various extracts to the positive control, it was discovered that all of the extracts demonstrated a lower inhibitory effect compared to acarbose. The statistical analysis revealed a significant difference, with a *p*-value of less than 0.001 for EAcE (M), EM (S), EAcE (S), and a *p*-value of less than 0.01 for ME (M), ME (S), and AQE. 

### 2.6. Molecular Modeling Studies

Alpha-amylase and alpha-glucosidase are enzymes that play a role in the digestion of carbohydrates. α-amylase breaks down starch into smaller carbohydrates, while α-glucosidase breaks down these smaller carbohydrates into simple sugars, such as glucose, which can be absorbed into the bloodstream. In individuals with diabetes, these enzymes can contribute to high blood sugar levels if glucose intake or storage are not adequately regulated. To manage blood sugar levels, some people with diabetes may take α-amylase or α-glucosidase inhibitors to slow down carbohydrate digestion and absorption. All molecules identified in the *G. bursa-pastoris* extracts using HPLC showed remarkable inhibitory activity against α-amylase and α-glucosidase compared with acarbose and metformin (positive controls), as shown in Figure 4 and Figure 5. Rutin showed the highest glide gscore values against α-amylase of −8.109 kcal/mol, followed by naringenin, with a glide gscore of −7.198 kcal/mol. Moreover, quercetin and kaempferol were the most active molecules in the active site of α-glucosidase, with a glide gscore of −7.035 and −5.698 kcal/mol, respectively (Table 6). 

In comparison to the effect of acarbose and metformin in the active site of α-amylase, apigenin, kaempferol, naringenin, quercetin, and rutin showed higher binding energy than acarbose (−7.067 kcal/mol) (Table 6).

However, the effect of acarbose and metformin in the active site of α-glucosidase, quercetin showed higher binding energy than acarbose (−6.989 kcal/mol), while 4-hydroxybenzoic acid, apigenin, kaempferol, luteolin, naringenin, quercetin, quercetin 3-glucoside, rutin, and vanillin exhibited higher binding energy than metformin.

## 3. Discussion

In this research, extracts were prepared from extracts of *G. bursa-pastoris.* The results showed that polar solvents (methanol and water) produced higher yields than less polar solvents (hexane). This is supported by published research [39,40]. Additionally, polar solvents such as water are capable of extracting phenolic compounds that are bound to sugars or proteins, saponins, glycosides, organic acids, selenium, and mucilage [41]. Furthermore, the study by Bandar et al. [42] concluded that Soxhlet extraction was the most suitable technique for extracting active molecules of *G. bursa-pastoris* based on yield.

For EAcE and ME of *G. bursa-pastoris*, the mean quantity of total polyphenols and total flavonoids was determined. The polyphenolic content of the methanolic extract of *G. bursa-pastoris* was investigated by Ramdani et al. in Morocco [43] and by Yildiz et al. in Turkey [44]. Ramdani et al. reported high phenolic content using maceration, with a value of 99.62 mg GAE/g, and using extraction at 40 °C, with a value of 142.26 mg GAE/g. In contrast, Yildiz et al. found a relatively low phenolic content of 0.35 mg GAE/g. This disparity in results could be attributed to differences in the origin of the seaweed sample, the extraction procedure used, the maturity of the algae, the growing conditions or the extraction solvent used [45,46].

The chemical composition of EAcE and HE was determined via gas chromatography coupled with mass spectrometry (GC-MS). It highlights the presence of palmitic acid, linoleic acid and eicosenoic acid as major compounds. Linoleic acid is an essential fatty acid and has hypocholesterolemic effects [47]. Palmitic acid inhibits prostate cancer cell proliferation and metastasis by suppressing the PI3K/Akt pathway [48]. Omega-3 fatty acids, particularly linolenic acid, play a crucial role in the regulation of blood glucose levels by enhancing insulin secretion from the beta cells within the pancreatic islets of Langerhans [49]. Linoleic acid enhances glucose uptake by C2C12 muscle cells [50]. Oleic acid has been found to promote insulin secretion in the glucose-sensitive INS-1 cell line. Of particular significance is its ability to enhance insulin secretion even when the inflammatory cytokine TNF-α is present [51]. Stearic acid enhances glucose uptake into adipocytes by activating insulin/insulin receptor signaling while inhibiting PTP1B [52]. The expression of the GLUT4 transporter and the facilitation of glucose uptake by 3T3-L1 adipocytes are significantly impacted by polyunsaturated fatty acids, resulting in an increased abundance of both GLUT4 and GLUT1 transporters [53,54]. In addition, the effect of eicosenoic acid, 7,10-hexadecadienoic acid, on diabetes is not well-studied compared to other omega fatty acids, such as omega-3 and omega-6 fatty acids.

It is important to consider that the type and proportion of fatty acids in a food can have a significant impact on its overall health effects. For instance, high levels of SFAs have been associated with an increased risk of cardiovascular disease, while PUFAs have been linked to a reduced risk. MUFAs, on the other hand, are commonly considered to be beneficial for health as they may help to lower cholesterol levels and decrease the risk of heart disease [55]. Polyunsaturated fatty acids, or PUFAs, play a crucial role in human metabolism [56], as they constitute a major part of cell membrane phospholipids [57]. Their metabolic significance is reflected in their diverse biological properties, such as antibacterial activity [58,59,60], anti-inflammatory effects [61,62], antioxidant capacity [63], potential for preventing cardiovascular disease [64], and inhibition of tumor growth [65,66]. 

The major phenolic molecules of the different EAcE and ME extracts were determined through HPLC analysis. The study revealed that *G. bursa-pastoris* extracts are an excellent source of flavonoids and polyphenols. The investigation of natural bioactive compounds possessing health-promoting properties has been the subject of substantial scrutiny. Among these compounds, caffeic acid is a hydroxycinnamic acid that is synthesized as a secondary metabolite by several plant species [67]. It is known for its antioxidant [68,69], anticancer [70] and anti-diabetic properties [71]. Caffeic acid has also been found to protect neurons against oxidative damage and neurotoxicity by modulating various signaling pathways and reducing inflammation in the brain [70].

7,3′,4′-flavon-3-ol, also known as fisetin, is a natural flavonoid that has been extensively studied in the scientific community due to its various health-promoting properties. This compound acts as an anti-inflammatory and anticancer agent [72,73,74] and has also been shown to have anti-diabetic properties that can improve insulin sensitivity and decrease blood sugar levels, making it potentially useful in managing Type 2 diabetes [75]. Furthermore, fisetin has demonstrated beneficial effects in preclinical models of several neurodegenerative diseases, including Alzheimer’s disease, vascular dementia, schizophrenia, Parkinson’s disease, amyotrophic lateral sclerosis, Huntington’s disease, stroke, depression, and traumatic brain injury [76].

The DPPH assay is predicated on the ability of the stable free radical DPPH (•) to diminish coloration in the presence of antioxidants. This radical features an unpaired electron that confers a deep purple coloration, which is diminished by the presence of antioxidants [77]. DPPH assay is widely applied as an indicator to assess the antioxidant activity of diverse plant extracts [78,79]. The β-carotene/linoleic acid bleaching test measures the impact of antioxidants on the discoloration of a solution caused by the oxidation of linoleic acid, which generates hydroperoxides that attack β-carotene molecules. Antioxidants, on the other hand, inhibit β-carotene bleaching [80]. The results showed that extracts of the red algae *G. bursa-pastoris* with high total phenol content demonstrated robust antioxidant activity, suggesting that algal polyphenols may be the key contributors to the radical scavenging properties of these extracts. There is a strong correlation between antioxidant activity and the content of polyphenols and flavonoids, as noted in the literature. These algae are, therefore, a rich source of bioactive compounds and have significant potential for antioxidant activity, as evidenced by increased efficacy resulting from heating during Soxhlet extraction. Ramdani et al., [43] reported a high antioxidant activity, with an IC_50_ of 0.09 to 0.1% for the ethanolic extract and 0.17 to 0.18% for the aqueous extract, the differences of which could be attributed to the extraction method, time, and temperature.

The in vitro anti-diabetic activities of *G. bursa-pastoris* extracts were investigated through their potential to inhibit pancreatic enzyme activity. Type 2 diabetes is widely recognized as the most prevalent metabolic disorder affecting people of all ages across the globe. This condition is known to cause numerous long-term health complications, such as retinopathy, nephropathy, neuropathy, cardiopathy, and, unfortunately, even death. In fact, the mortality rate associated with this condition has seen a concerning increase of 3% between the years 2000 and 2019 [30]. Therefore, it is crucial to have proper medical supervision, which involves the use of oral anti-diabetic medications such as sulfonamides, biguanides, and metformin [81]. As the disease progresses, subcutaneous insulin injections become a common method of treatment. While these injections can be helpful in regulating blood sugar levels and promoting bodily balance, they also carry a number of adverse effects, including hypoglycemia, coma, gastrointestinal issues, diarrhea, and even heart failure, nausea, and vomiting [33]. As such, it has become imperative to conduct research on natural products with anti-diabetic properties. When it comes to human nutrition, starch serves as the primary source of carbohydrates that can be digested, and it undergoes a breakdown process into glucose as a result of digestive enzymes such as α-amylase. This enzymatic reaction is considered to be the initial stage of starch digestion [82]. This particular enzyme is capable of being secreted by both the salivary glands and the pancreas and then released into the intestine. Its primary function is to hydrolyze the 1,4-glucan bonds of polysaccharides into oligosaccharides and disaccharides [83]. As a means of controlling postprandial hyperglycemia, natural α-amylase inhibitors are considered to be one of the best options available [84]. In the present work, we examined the inhibitory effects of different *G. bursa-pastoris* extracts on pancreatic α-amylase and α-glucosidase in vitro. Our results showed that the various extracts possessed considerable in vitro inhibitory α-amylase activity, particularly the methanolic extract, which yielded an IC_50_ value that was comparable to that of the positive control, acarbose. This was followed by the aqueous extract and ethyl acetate extract. *G. bursa-pastoris* was found to contain a total of 0.35 mg of gallic acid per gram of phenolic compounds [44]. In other studies, this chemical has been shown to have significant anti-diabetic activity and has the ability to block digestive enzymes linked to diabetes, such as pancreatic α-amylase [85,86]. Our samples were found to contain significant levels of gallic acid, apigenin, and tannic acid, which have all been shown to exhibit intense α-amylase activity by many researchers. Tannic acid has an IC_50_ value of 0.75 mg/mL, compared to acarbose’s 0.01 mg/mL. Gallic acid has an IC_50_ value of 287.53 mg/mL, compared to acarbose’s 678.43 mg/mL. Apigenin has an IC_50_ value of 3.46 mg/mL, compared to acarbose [87,88,89]. In addition, it was found that gallic acid can enhance insulin secretion by promoting peripheral glucose uptake by adipocytes and skeletal muscle [90]. Apigenin, on the other hand, plays a role in decreasing hepatic glucose-6-phosphatase activity that catalyzes the final reaction of glycogenolysis and glucose release into the bloodstream [91]. In diabetic mice, kaempferol was observed to contribute to the improvement of hyperglycemia and reduce hepatic glucose production by decreasing the activity of pyruvate carboxylase [92]. Quercetin exhibits anti-diabetic properties, including reducing postprandial glycemia, achieved by inhibiting glucose uptake by skeletal muscle [93]. Chlorogenic acid possesses potential effects on diabetes. It has been found to regulate blood sugar levels and enhance insulin sensitivity [94]. Syringic acid, flavanone, and naringin have been reported to have antihyperglycemic properties, which can lead to reduced glucose absorption and improved glycemic control [95,96,97]. One of the organic acids identified in our algae extracts is succinic acid. It is present in an amount of 3.01% for ME (M) and has a high antihyperglycemic effect (IC_50_ = 6.74 μM/mL, compared to acarbose’s 0.21 μM/mL). According to Gamze Yildiz et al., [44], *G. bursa-pastoris* contains a high amount of ascorbic acid, with a value of 21.6 mg/100 g, which has been confirmed in our work. This compound has been shown to have the ability to reduce blood glucose, enhance insulin secretion, improve insulin resistance, and reduce diabetic complications [98]. The same study also showed that *G. bursa-pastoris* [EHL1] contains vitamin E, with an amount of 57 mg α-tocopherol/100 g, which has a potential anti-diabetic effect. Tocopherols play a key role in improving insulin resistance and suppressing oxidative stress, both of which are implicated in diabetes management [99]. 

On the other hand, molecular docking simulations of the metabolites identified in *G. bursa-pastoris* extracts were performed. Several scientific studies have indicated that rutin plays a significant role in terms of anti-diabetic activity by enhancing glucose homeostasis in diabetic rats. It achieves this by lowering fasting blood glucose levels and increasing insulin levels, as well as by increasing glycogen content in the liver and muscle while reducing it in the kidneys. Rutin also boosts hexokinase activity and reduces glucose-6-phosphatase and fructose1,6-bisphosphatase activities in the tissues [100,101].

Another study showed that a fraction rich in rutin, naringenin, and quercetin from *Globularia alypum* exhibited significant anti-diabetic activity with an inhibitory effect on α-amylase (IC_50_ = 0.57 mg/mL) and α-glucosidase (IC_50_ = 0.52 mg/mL) [102]. In comparison, another in silico study showed significant affinity of rutin and quercetin against alpha-amylase and alpha-glucosidase [103]. *G. bursa-pastoris* is rich in polyphenols, such as flavonoids (quercetin, kaempferol, catechins) and phenolic acids (chlorogenic acid, caffeic acid) that have been reported to possess alpha-amylase and alpha-glucosidase inhibitory effects [104]. Furthermore, polyunsaturated fatty acids have also been studied for their potential effects on alpha-amylase and alpha-glucosidase inhibitory activities [105]. Some studies have suggested that omega-3 fatty acids such as linolenic acid and omega-6 fatty acids such as linoleic acid, present in *G. bursa-pastoris*, may have inhibitory effects on these enzymes, potentially contributing to their anti-diabetic properties [106].

Molecular docking of rutin in the active sites of α-amylase showed that rutin formed eight hydrogen bonds with amino acid residues TRP 59, THR 163, GLN 63, ASP 197, GLU 233, LYS 200, and HIS 201, while naringenin established in this active site of alpha-amylase two hydrogen bonds with residues ASP 197 and GLN 63 and 2 Pi-Pi stacking bond with amino residues TYR 62 and TRP 59. Furthermore, quercetin established in the active site of alpha-glucosidase four hydrogen bonds with residues ASP 518, SER 676 ASP 616, and two Pi-Pi staking bonds with residue PHE 649, whereas kaempferol established in this active site of alpha-glucosidase three hydrogen bonds with residues ASP 518, ASP 616, and SER 676 and one Pi-Pi stacking with residue PHE 649.

## 4. Materials and Methods

### 4.1. Chemicals and Reagents

For the purposes of determining the total phenolic and flavonoid components, analytical-grade chemicals and reagents were procured from Merck (Darmstadt, Germany) and Carl Roth GmbH (Karlsruhe, Germany). N-hexane, ethyl acetate, and ethanol were obtained from Merck (Darmstadt, Germany). The required acarbose, 3,5-Dinitrosalicylic acid (DNSA), 1,1-Diphenyl-2-picrylhydrazyl (DPPH•), β-carotene, α-glucosidase, and α-amylase were purchased from Merck (Sigma-Aldrich, St. Louis, MA, USA). Ascorbic acid, kojic acid, gallic acid, apigenin, succinic acid, cholesterol, and tannic acid, which served as phenolic standards, were obtained from Merck and Carl Roth GmbH (Karlsruhe, Germany).

### 4.2. Plant Material and Extraction

The red algae *G. bursa-pastoris* was harvested from the Nador lagoon (35°08′26.9″ N 2°29′09.6″ W) in northern Morocco in April 2021, as shown in Figure 6. The algae species was classified by Dr. M. RAMDANI, who is affiliated with the Faculty of Sciences at University Mohammed I in Oujda, Morocco.

The collected algae sample was brought to the laboratory for processing. *G. bursa-pastoris* was thoroughly cleaned and repeatedly rinsed with distilled water before being dried and left to be exposed to light for 48 h. The dried sample was then placed in an oven at 35 °C for 24 h (as shown in Figure 7). The seaweed was lyophilized and ground into a powder for extraction purposes. The extraction process utilized maceration and Soxhlet techniques, using four solvents: hexane, ethyl acetate, methanol, and distilled water. The resulting extracts were placed in flasks and filtered using a glass filter crucible (50 mL, with Porosity 4, Isolab, Wertheim, Germany). The extracts were then dehydrated using a rotary evaporator (Laborota 4000, Heidolph Instruments, Schwabach, Germany). The extraction yield was calculated using the following equation:$$\mathrm{Yield}\;(\%)=\frac{mass\;dried\;extract\;(g)}{mass\;dried\;matrix\;(g)}\;\times \;100$$

The data were recorded as the median of three extraction replicates.

#### 4.2.1. Maceration Extraction

This extraction method entails immersing a solid substance in a cold liquid to draw out soluble compounds. The extracts were prepared by mixing 100 g of macroalgae powder with 200 mL of solvent (99% n-hexane (2 h), ethyl acetate (24 h), methanol (24 h), and distilled water (24 h)) at ambient temperature, through stirring (as shown in Figure 8).

#### 4.2.2. Soxhlet Extraction

The Soxhlet extraction method requires the use of a condenser, a Soxhlet chamber, and an extraction flask. Then, 35 g of powder from marine macroalgae species was placed in an extraction thimble and combined with 300 mL of the selected solvent (hexane, ethyl acetate, or methanol) in the extraction flask. The duration of the Soxhlet extraction process was determined based on the time required for the descending solvent to become colorless.

### 4.3. Phytochemicals Compounds

#### 4.3.1. Quantification of Total Phenolic Constituents

The total polyphenol content in *G. bursa-pastoris* extracts was determined using a modified Folin–Ciocalteu method. Indeed, 200 μL of extract solution with a concentration of 1 mg/mL was mixed with 1000 μL of Folin–Ciocalteu reagent solubilized in distilled water, and finally, 800 μL of sodium carbonate was added (75 g/L); the resulting mixture was incubated at room temperature for 1 h in the dark. Then, the absorbance was measured at 765 nm using a spectrophotometer against ethanol as a control. The calibration curve is created by varying the concentration of gallic acid (0–0.1 mg/mL). All experiments were performed in triplicate to take the mean of the experiments with the standard deviation. The amount of total phenolic compounds was expressed in mg gallic acid equivalents per gram of dry extract (mg GAE/g) [107].

#### 4.3.2. Measurement of Total Flavonoids Contents

A modified version of the method described by Kim et al. [108] was used. A quantity of 200 µL of each extract of *G. bursa-pastoris*, 1000 µL of distilled water and 50 µL of NaNO_2_ (5%) were combined. Following a 6 min incubation period, 120 µL of AlCl_3_ (10%) was introduced, followed by a 5 min incubation period at room temperature in the dark. Then, 400 µL of 1 M NaOH and 230 µL of distilled water were mixed together. At 510 nm, the absorbance was calculated. The calibration curve was created by using different concentrations of quercetin solution (0–0.1 mg/mL) as a standard. All measurements were determined three times to verify the reproducibility of the results.

### 4.4. Fatty Acid GC-MS Analysis of G. bursa-pastoris Extracts

In accordance with the protocol described in Loukili et al. [109] with some modifications, methyl esters of hexane fatty acids and ethyl acetate were extracted from *G. bursa-pastoris*. The characterization and identification of fatty acids were analyzed using the Shimadzu GC system (Kyoto, Japan), which was equipped with a BPX25 capillary column with a 5% diphenyl and 95% dimethylpolysiloxane phase (30 m × 0.25 mm × 0.25 µm) and connected to a QP2010 MS. Helium gas (99.99%) was used as the mobile phase with a flow rate of 3 L/min. The temperature of the injection, ion source, and interface was set at 250 °C, while the temperature program for the column oven was set to 50 °C for 1 min and then heated to 250 °C at a rate of 10 °C/min and maintained for an additional minute. The ionization of the sample components was performed in Electron Ionization (EI) mode at 70 eV, with a scanned mass range of 40 to 300 m/z. A 1 μL volume of each extract was injected in split mode, and each sample was analyzed in triplicate. The compounds were characterized by comparing their retention times, mass spectra fragmentation patterns, and databases, including the National Institute of Standards and Technology’s database (NIST). Data processing was carried out using LabSolutions (version 2.5, Shimadzu, Kyoto, Japan).

### 4.5. HPLC Analyses of G. bursa-pastoris Extracts

The phenolic compounds in the ethyl acetate and methanolic extracts were identified using High-Performance Liquid Chromatography (HPLC) with an Agilent 1100 system (Agilent Technologies, Palo Alto, CA, USA) connected to a diode array UV detector (Bruker, Germany). Each extract (10 µL) was passed through a Zorbax XDB-C18 column (5 m, 250 × 4.6 mm) attached to the Agilent 1100 system, preceded by a 4 × 3 mm C18 cartridge precolumn (Agilent Technologies). The elution gradient used was 0 to 5 min: 95% A and 5% B, 25 to 30 min: 65% A and 35% B, 35 to 40 min: 30% A and 70% B, and 40 to 45 min: 95% A and 5% B. The elution system comprised A (water/MeOH (9/1) + 0.1% phosphoric acid) and B (methanol + 0.1% phosphoric acid), with a constant flow rate of 1 mL/min at a temperature of 40 °C. Spectrophotometric measurements were taken at wavelengths of 254, 280, 320, 370, and 510 nm. Compounds were identified by comparing their retention times and UV spectra to those of standards.

### 4.6. Antioxidant Activity

#### 4.6.1. Scavenging 2, 2-Diphenyl-1-Picrylhydrazyl Radical Test

The antioxidant activity of various extracts of *G. bursa-pastoris* was evaluated using the 1,1-diphenyl-2-picrylhydrazyl (DPPH) radical bleaching technique, as described by Brand-Williams et al. with slight modifications [110]. The initial concentration of extracts and Ascorbic acid employed is 1 mg/mL. A mixture of 0.8 mL of samples or standard (ascorbic acid) of varying concentrations (ranging from 0.02 to 0.32 mg/mL) and 2 mL of DPPH• solution (2 mg of DPPH• in 200 mL MeOH) was prepared and shaken by hand. After 30 min of incubation in the dark at ambient temperature, the absorbance of the samples was measured using a UV/visible spectrophotometer at a wavelength of 517 nm, with respect to the blank. Each determination was performed in triplicate. 

The inhibitory activity of the DPPH radical by a sample was calculated using the following equation:$$\mathrm{Inhibition}\;\mathrm{Percent}\;=\;[\frac{\left(A_{b}-A_{s}\right)}{A_{b}}]\;\times \;100$$
where A_b_: Absorbance of the blank, A_s_: Absorbance of sample (or positive control).

The graph plotting inhibition percentage against extract concentration was used to calculate the IC_50_.

#### 4.6.2. β-Carotene Bleaching Assay

The antioxidant activity of *G. bursa-pastoris* extracts was determined using the method described by Bekkouch et al. [111]. This evaluation was based on the extracts’ ability to reduce oxidative damage to β-carotene in an emulsion using the carotene bleaching test. To prepare the emulsion, 2 mg of β-carotene was dissolved in 10 mL of chloroform, along with 20 mg of linoleic acid and 200 mg of the emulsifier Tween 80. The chloroform was then evaporated at 40 °C using a vacuum evaporator, and 100 mL of distilled water was added to the solution while stirring vigorously. 0.2 mL of the emulsion was placed in separate test tubes, along with either the extract or a reference antioxidant solution (BHA) at a concentration of 1 mg/mL. Absorbance was measured at 470 nm using a 96-well microplate reader at t0, immediately after adding the emulsion and after 2 h. All measurements were performed in triplicate. The inhibition of the linoleate/β-carotene radical was calculated using the following formula:$$\mathrm{Bleaching}\;\mathrm{inhibition}\;(\%)\;=\;100\;-\;\frac{initial(\beta -carotene)(t0)-(\beta -carotene)\;after\;2\;h}{initial(\beta -carotene)(t0)}$$

### 4.7. In Vitro α-Amylase Inhibition

The inhibitory effect of the different extracts of *G. bursa-pastoris* on α-amylase was determined according to the method described by Daoudi et al. [84]. A reaction mixture was prepared containing 0.2 mL of the sample or acarbose (used as a positive control at varying concentrations ranging from 2.27 to 0.14 mg/mL), 0.2 mL of phosphate buffer (pH 6.9), and 0.2 mL of enzyme solution (13 IU). The mixture was pre-incubated at 37 °C for 10 min before the addition of 0.2 mL of enzyme–substrate (1% starch dissolved in phosphate buffer). The reaction was allowed to continue at 37 °C for 30 min. The enzymatic reaction was then stopped by adding 0.6 mL of DNSA reagent, followed by incubation at 100 °C for 8 min and placement in a cold-water bath. The absorbance was measured at 540 nm. 

The following formula was employed to calculate the inhibition percentage:$$Inhibition\;activity\left(\%\right)=\frac{\left(OD\;control\;540\;nm-OD\;control\;blank\;540\;nm\right)-(OD\;sample\;540\;nm-OD\;sample\;blank\;540\;nm)}{OD\;control\;540\;nm-OD\;control\;blank\;540\;nm}\times 10$$

The IC_50_ of the various test was presented graphically using the function:$$Inhibition\;percentage\left(\%\right)=f(\log\left(sample\;concentration\right))$$

### 4.8. In Vitro α-Glucosidase Inhibition Assay

The impact of *G. bursa-pastoris* extracts on intestinal α-glucosidase activity was determined through a modified version of the protocol described by Hbika et al. [112]. A mixture was prepared by combining 100 mL of sucrose (50 mM), 1000 mL of phosphate buffer (50 mM, pH 7.5), and 100 mL of intestinal α-glucosidase enzyme solution (10 IU). This mixture was then added to 10 mL of distilled water (as control), acarbose (as positive control), or *G. bursa-pastoris* extract solution at a concentration of 2.2 g/mL. The mixture was then incubated for 25 min at 37 °C in a water bath. To inhibit the enzymatic reaction and measure the release of glucose, the mixture was heated at 100 °C for 5 min:$$Inhibition\;activity\left(\%\right)=\frac{\left(OD\;control\;540\;nm-OD\;control\;blank\;540\;nm\right)-(OD\;sample\;540\;nm-OD\;sample\;blank\;540\;nm)}{OD\;control\;540\;nm-OD\;control\;blank\;540\;nm}\times 10$$

### 4.9. Molecular Docking Study

The ligands (compounds) identified via HPLC analysis in *Gracilaria bursa-pastoris* were obtained from PUBCHEM.SDF format, then processed for docking using the LigPrep tool in Schrödinger Software Maestro 11.5 using the OPLS3 force field. The ionization states at pH 7.0 ± 2.0 were selected, resulting in a maximum of 32 stereoisomers for each ligand. The protein preparation process involved downloading the 3D crystal structure of α-amylase (PDB: 1B2Y) and α-glucosidase (PDB: 5NN8) in PDB format from the protein data bank [113]. The structure was then refined using the Schrödinger-Maestro v11.5’s Protein Preparation Wizard. This included assigning charges and bond orders, adding hydrogens to heavy atoms, converting selenomethionines to methionines, and removing all water molecules. Minimization was performed using the OPLS3 force field with a maximum heavy atom RMSD limit of 0.30 Å [114].

The generation process is initiated by selecting a ligand atom, leading to the preparation of a default grid box. The grid box has a 20 × 20 × 20 volumetric spacing, with its coordinates being x: 18.938, y: 5.742, and z: 46.879 for α-amylase and the following coordinates x: −14.02, y: −38.177, and z: 95.206 for α-glucosidase. The ligand was then linked to the protein-generated grid box through “Extra Precision” (XP). The outcome was assessed based on the XP GScore. The Glide Standard Precision (SP) Ligand Docking was performed using Schrödinger-Maestro v11.5. In this process, penalties were imposed on non-cis/trans amide bonds, and the van der Waals scaling factor was set to 0.80, with a partial charge cutoff of 0.15 for the ligand atoms. The final scoring was based on the energy-minimized poses displayed as Glide scores. The ligand with the lowest Glide score was recorded as the best-docked pose.

## 5. Conclusions

The abundance of *G. bursa-pastoris* in the Marchica Nador lagoon sparked our curiosity about potential uses for these natural products. Upon conducting GC-MS analysis, we discovered that both hexane and ethyl acetate extracts from *G. bursa-pastoris* contained a high proportion of unsaturated fatty acids, with palmitic acid being the primary compound detected in three extracts (HE (M), EAcE (M), and EAcE (S)) at levels ranging from 38% to 50%. The hexane extract obtained through maceration had the highest amount of eicosenoic acid, while apigenin was the main compound in the ethyl acetate and methanolic extracts. Although the aqueous extract exhibited moderate antioxidant activity using the DPPH radical scavenging method, it was lower than that of ascorbic acid. However, these extracts, containing numerous compounds, could be an alternative to synthetic additives, and it is likely that once purified, they could exhibit comparable activity to ascorbic acid. The methanolic extract was particularly noteworthy, displaying excellent antioxidant activity in the β-carotene bleaching test, with an IC_50_ (0.062 ± 0.64 mg/mL) closely resembling BHA when it was used as a reference. Moreover, our study found that *G. bursa-pastoris* extracts exhibited a significant anti-diabetic effect in both in vitro and in silico assays, suggesting the potential as a phytomedicine to regulate blood sugar levels and prevent health complications in diabetic patients, as well as in healthy subjects.

## Figures and Tables

**Figure 1 marinedrugs-21-00372-f001:**
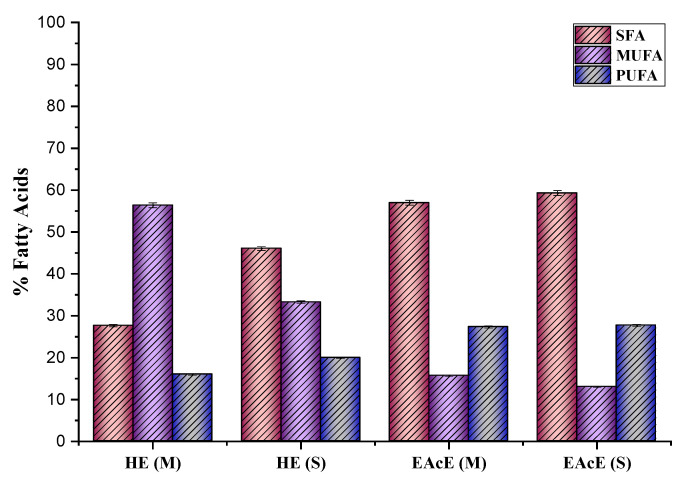
Fatty acid (FA) content in different extracts of *G. bursa-pastoris* (SFAs: saturated fatty acids; MUFAs: monounsaturated fatty acids; PUFAs: polyunsaturated fatty acids; HE: hexane extract; EAcE: ethyl acetate extract; M: maceration; S: Soxhlet extraction).

**Figure 2 marinedrugs-21-00372-f002:**
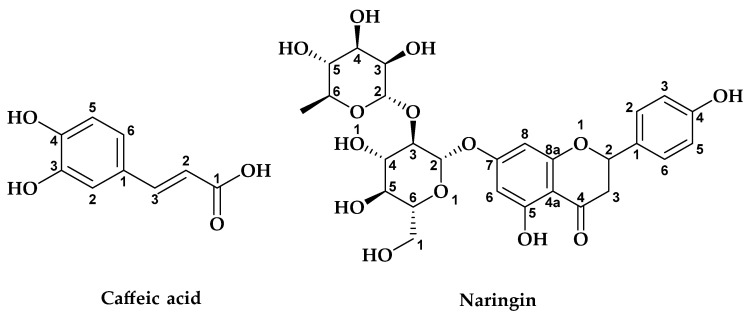
Structures of the main compounds present in the red algae *G. bursa-pastoris*.

**Figure 3 marinedrugs-21-00372-f003:**
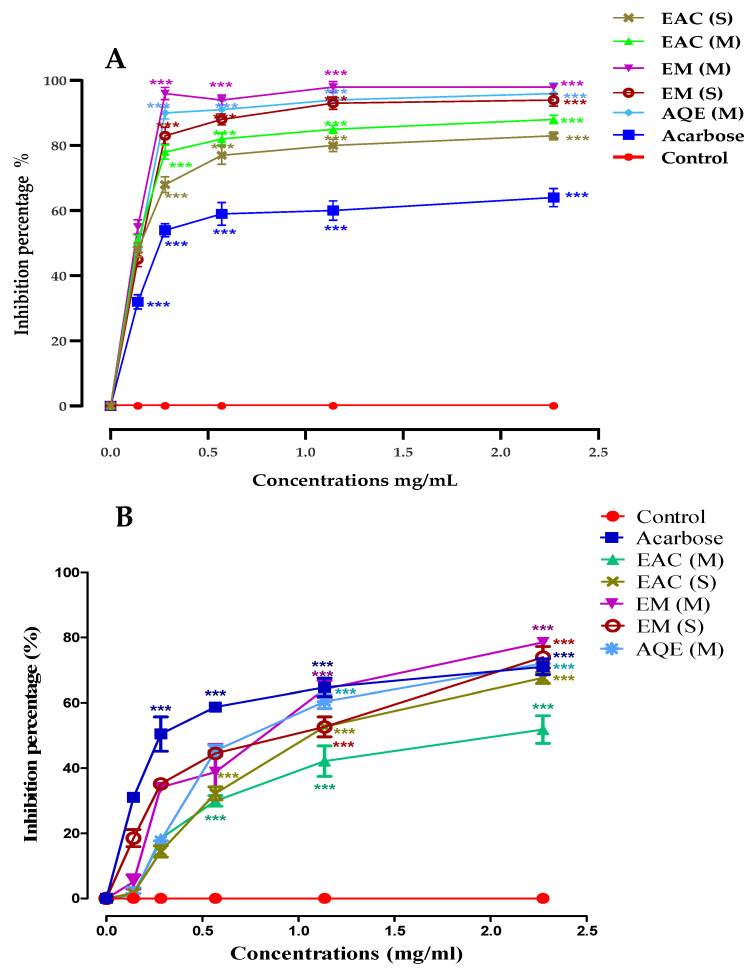
Inhibition percentage of *G. bursa-pastoris* extracts and acarbose against α-glucosidase (**A**) and α-amylase (**B**) at different doses of *G. bursa-pastoris extracts* EAc: Ethyl Acetate; ME: Methanolic Extract; AQE: Aqueous Extract; M: Maceration; S: Soxhlet. Data are expressed as mean ± SD. *** *p* < 0.001.

**Figure 4 marinedrugs-21-00372-f004:**
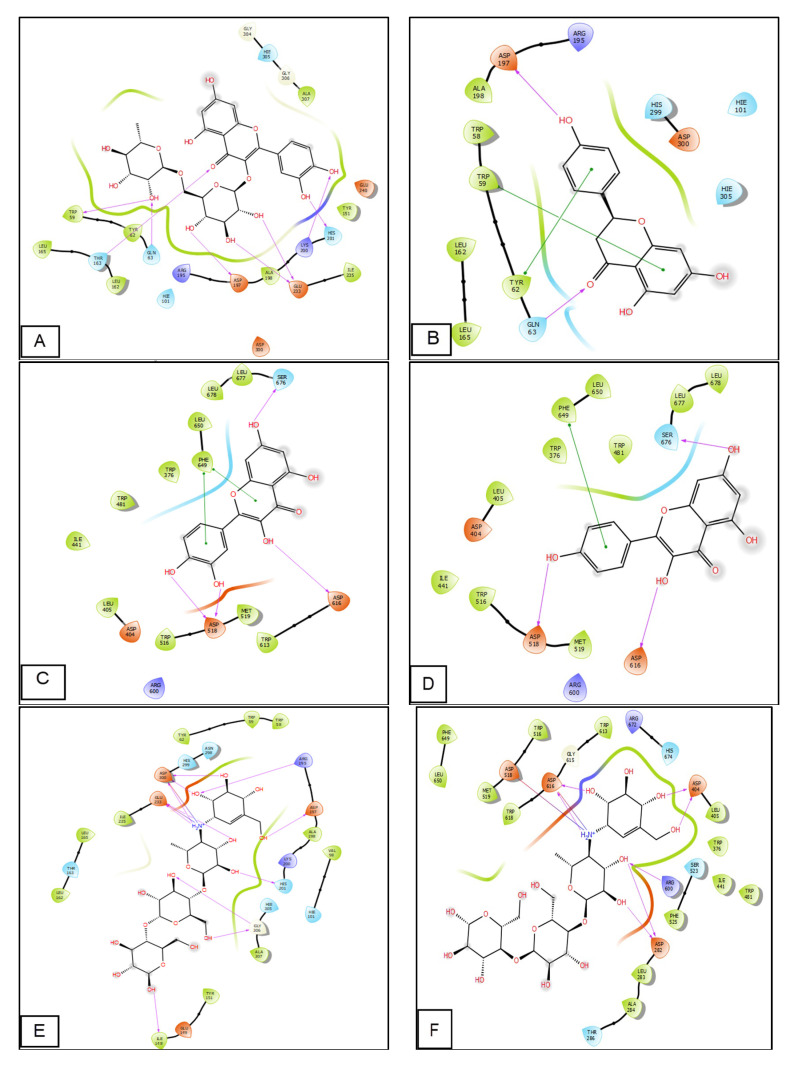
Two-dimensional view of ligands interactions. (**A**): Rutin interactions in the active sites of α-amylase; (**B**): naringenin interactions in the active sites of α-amylase; (**C**): quercetin interactions in the active site of α-glucosidase; (**D**): kaempferol interactions in the active site of α-glucosidase; (**E**,**F**): acarbose interactions in the active site of α-amylase and α-glucosidase; (**G**,**H**) metformin interactions in the active site of α-amylase and α-glucosidase.

**Figure 5 marinedrugs-21-00372-f005:**
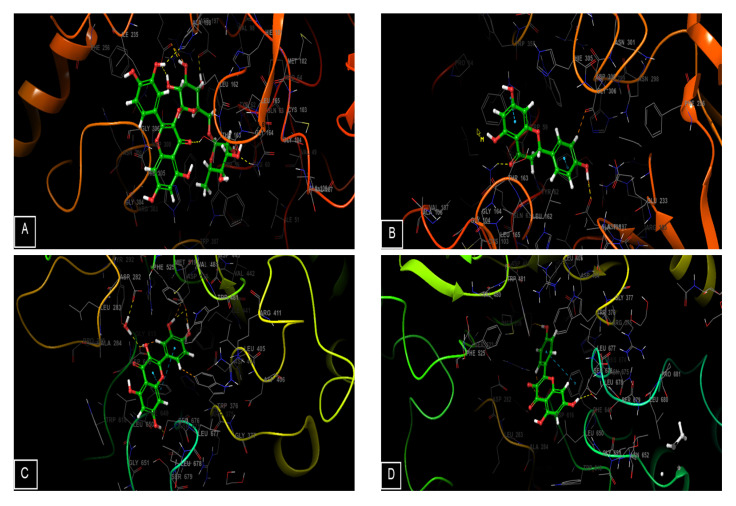
Three-dimensional view of ligand interactions. (**A**): rutin interactions in the active sites of α-amylase; (**B**): naringenin interactions in the active sites of α-amylase; (**C**): quercetin interactions in the active site of α-glucosidase; (**D**): kaempferol interactions in the active site of α-glucosidase; (**E**,**F**): acarbose interactions in the active site of α-amylase and α-glucosidase; (**G**,**H**): metformin interactions in the active site of α-amylase and α-glucosidase.

**Figure 6 marinedrugs-21-00372-f006:**
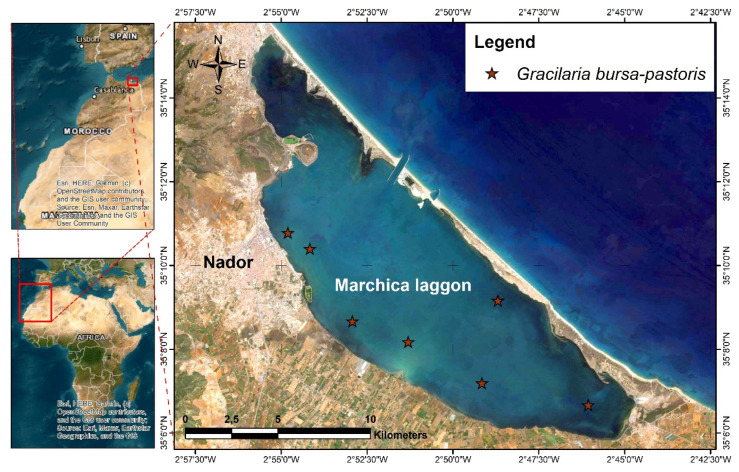
Location of the *G. bursa-pastoris* sampling site in the Nador lagoon in the Mediterranean Sea.

**Figure 7 marinedrugs-21-00372-f007:**
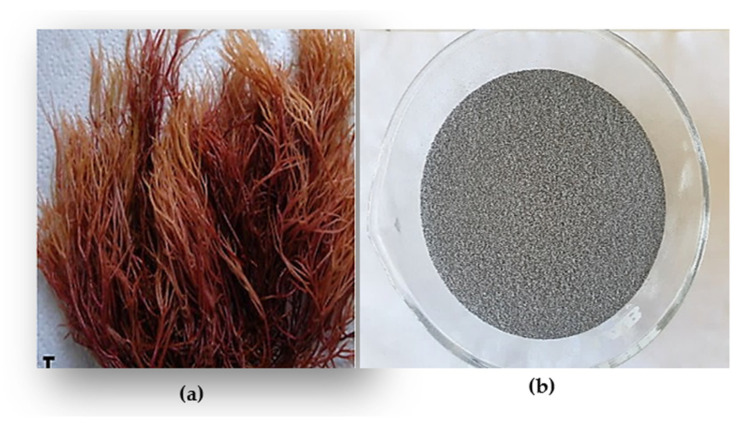
(**a**) *G. bursa-pastoris*, (**b**) powder of dried *G. bursa-pastoris*.

**Figure 8 marinedrugs-21-00372-f008:**
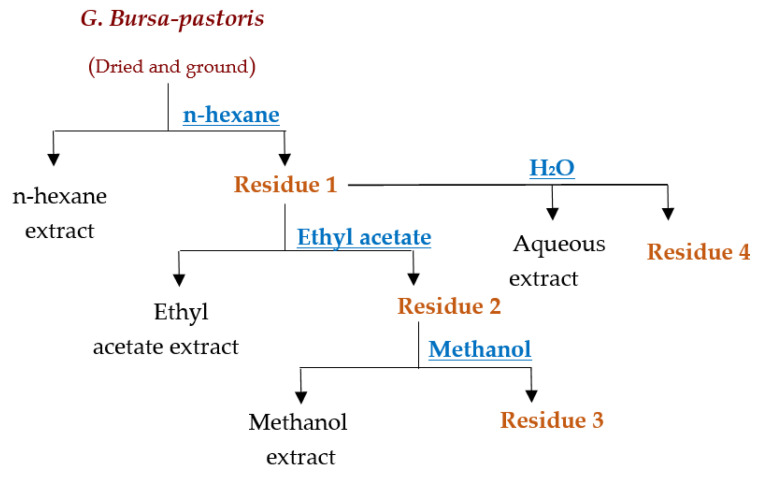
Extraction process of *G. Bursa-pastoris* with various solvents.

**Table 1 marinedrugs-21-00372-t001:** Yields, phenols and flavonoids contents of different extracts of *G. bursa-pastoris*.

Solvent	Extraction Methods	Yield (%)	Polyphenols (mg GAE/g)	Flavonoids (mg QE/g)
Hexane	M	0.11 ± 0.09	-	-
	S	1.16 ± 0.34	-	-
Ethyl acetate	M	0.21 ± 0.04	165.42 ± 0.21	84.53 ± 0.07
	S	1.80 ± 0.08	150.24 ± 0.11	75.47 ± 0.02
Methanol	M	0.81 ± 0.26	37.69 ± 1.02	15.29 ± 0.18
	S	3.42 ± 0.07	28.68 ± 0.07	13.70 ± 0.03
Water	M	1.25 ± 0.06	381.31 ± 0.33	201.80 ± 0.21

GAE: gallic acid equivalent; QE: quercetin equivalent; M: maceration extraction; S: Soxhlet extraction.

**Table 2 marinedrugs-21-00372-t002:** Fatty acids composition of hexane and ethyl acetate extracts from red algae *G. bursa-pastoris*.

Fatty Acids	RT (min)	HE (%)		EAcE (%)	
		M	S	M	S
Eicosenoic acid (C20:1)	20.08	53.06 ± 0.05	33.25 ± 0.41	5.21 ± 0.01	nd
7,10-Hexadecadienoic acid (C16:2)	21.17	2.88 ± 0.23	5.40 ± 0.01	5.67 ± 0.03	3.84 ± 0.06
Palmitic acid (C16:0)	23.30	22.88 ± 0.01	40.72 ± 0.31	50.55 ± 0.05	38.24 ± 0.02
Margaric acid (C17:0)	23.87	nd	nd	nd	16.19 ± 0.07
Oleic acid (C18:1)	24.54	3.32 ± 0.07	nd	10.48 ± 0.02	13.01 ± 0.09
Linoleic acid (C18:2)	25.03	9.51 ± 0.31	10.08 ± 0.11	15.57 ± 0.23	23.85 ± 0.11
Linolenic acid (C18:3)	25.09	3.58 ± 0.04	4.49 ± 0.17	6.09 ± 0.06	nd
Stearic acid (C18:0)	25.26	4.77 ± 0.43	6.06 ± 0.08	6.43 ± 0.07	4.87 ± 0.04
SFA ^a^		27.65	46.78	56.98	59.3
UFA ^b^		72.35	53.22	43.02	40.7
UFA/SFA ^c^		2.61	1.13	0.75	0.68

RT: retention time; M: maceration; S: Soxhlet extraction, HE: hexane extract; EAcE: ethyl acetate extract; nd: not detected; ^a^: saturated fatty acids (SFA); ^b^: unsaturated fatty acids (UFA); ^c^: unsaturation ratio = UFA/SFA.

**Table 3 marinedrugs-21-00372-t003:** Chemical composition of ethyl acetate and methanolic extracts from red algae *G. bursa-pastoris*.

N°	Compounds	RT (min)	EAcE (%)		ME (%)	
			M	S	M	S
1	Gallic acid	15.47	nd	nd	nd	0.637
2	Catechin	18.68	1.516	8.188	nd	nd
3	4-hydroxy benzoïc acid	18.91	4.105	18.778	nd	3.561
4	Chlorogenic acid	19.15	1.096	4.202	11.666	7.714
5	Caffeic acid	19.45	nd	nd	35.642	24.239
6	Syringic acid	19.74	nd	3.092	3.673	nd
7	Vanilline	23.1	3.457	nd	nd	nd
8	p-Coumaric acid	23.63	nd	nd	nd	5.269
9	Sinapic acid	24.09	9.534	nd	nd	nd
11	Quercetin-*O*-3-glucoside	24.52	8.074	nd	nd	nd
12	7,3′,4′-flavon-3-ol	24.92	8.316	20.682	nd	nd
13	Naringin	25.01	8.344	17.754	21.069	12.626
14	Rutin	25.16	8.291	nd	nd	nd
15	Salicylic acid	25.32	nd	15.613	14.901	11.854
16	Quercetin	25.46	8.669	nd	nd	nd
17	Cinnamic acid	25.48	8.941	nd	nd	9.662
18	Luteolin	25.64	8.912	nd	nd	9.387
19	Apigenin	25.87	7.839	nd	nd	nd
20	Kaempferol	26.1	9.108	11.690	8.100	9.109
21	Flavone	26.92	nd	nd	4.948	5.940
22	Flavanone	27.412	nd	3.795	nd	nd

RT: retention time; M: maceration; S: Soxhlet extraction; EAcE: ethyl acetate extract; ME: methanolic extract; nd: not detected.

**Table 4 marinedrugs-21-00372-t004:** IC_50_ values of *G. bursa-pastoris* extracts.

Extracts		IC_50_ (mg/mL)	
		DPPH	β-Carotene
EAcE	M	0.40 ± 0.51	0.83 ± 0.32
	S	0.30 ± 0.31	0.65 ± 0.04
ME	M	0.55 ± 0.25	0.07 ± 0.06
	S	0.46 ± 0.36	0.06 ± 0.64
AQE	M	0.17 ± 0.07	0.21 ± 0.14
Ascorbic Acid		0.06	-
BHA		-	0.02

EAcE: ethyl Acetate extract; ME: methanolic extract; AQE: aqueous extract; M: maceration; S: Soxhlet.

**Table 5 marinedrugs-21-00372-t005:** IC_50_ values of *G. bursa-pastoris* extracts and acarbose in α-amylase and α-glucosidase inhibition.

Inhibitors	IC_50_ (mg/mL)		
	α-Amylase		α-Glucosidase
Acarbose		0.35 ± 0.08	0.39 ± 0.04
EAcE	M	1.86 ± 0.06 ***	0.44 ± 0.02 ns
	S	1.11 ± 0.02 ***	0.57 ± 0.03 **
ME	M	0.72 ± 0.04 **	0.25 ± 0.09 *
	S	0.76 ± 0.05 **	0.37 ± 0.06 ns
AQE	M	0.85 ± 0.01 ***	0.32 ± 0.07 ns

EAcE: ethyl acetate extract; ME: methanolic extract; AQE: aqueous extract; M: maceration; S: Soxhlet, Data are expressed as mean ± SD. ns: non-significant * *p* < 0.05; ** *p* < 0.01; *** *p* < 0.001.

**Table 6 marinedrugs-21-00372-t006:** Docking results with identified compounds in active sites of α-amylase and α-glucosidase.

	α-Amylase (PDB: 1B2Y)			α-Gluosidase (PDB: 5NN8)		
	Glide Gscore	Glide Emodel	Glide Energy	Glide Gscore	Glide Emodel	Glide Energy
4-Hydroxybenzoic acid	−6.079	−27.887	−18.583	−4.366	−20.091	−13.882
7,3′,4′-flavon-3-ol	−4.801	−48.996	−38.542	−3.695	−36.338	−30.331
Apigenin	−7.105	−60.626	−41.637	−5.2	−44.196	−33.441
Caffeic Acid	−5.893	−39.405	−29.08	−4.237	−26.17	−18.858
Chlorogenic acid	−6.801	−63.522	−45.48	−3.738	−43.188	−37.071
Cinnamic acid	−3.632	−22.468	−18.689	−3.353	−19.589	−14.942
Flavanone	−5.798	−41.242	−30.237	−4.287	−35.284	−27.649
Flavone	−6.121	−43.458	−31.445	−4.326	−35.845	−28.673
Kaempferol	−7.2	−63.076	−43.51	−5.698	−47.8	−35.946
Luteolin	−6.366	−52.124	−38.738	−5.425	−51.621	−37.184
Naringenin	−7.198	−60.107	−40.882	−5.246	−38.138	−29.883
*p*-coumaric acid	−5.395	−35.602	−26.441	−3.558	−21.069	−15.352
Quercetin	−7.146	−67.907	−45.267	−7.035	−60.46	−42.363
Quercetin 3-glucoside	−6.402	−67.452	−49.322	−5.242	−60.877	−49.104
Rutin	−8.109	−108.772	−71.453	−5.237	−73.009	−57.522
Salicylic Acid	−4.06	−23.996	−18.556	−3.911	−15.12	−12.243
Sinapic acid	−3.894	−30.39	−24.001	−3.543	−27.834	−22.072
Syringic acid	−5.104	−32.584	−24.351	−3.472	−23.213	−19.151
Vanillin	−5.498	−24.397	−17.121	−4.651	−30.76	−23.024
Acarbose	−7.067	−108.733	−73.113	−6.989	−111.624	−70.347
Metformin	−3.989	−35.344	−22.492	−4.494	−36.414	−21.184

## Data Availability

Not applicable.

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
