# Peer review of "Study of the Phytochemical Composition, Antioxidant Properties, and In Vitro Anti-Diabetic Efficacy of Gracilaria bursa-pastoris Extracts"

_marinedrugs, 2023, doi:10.3390/md21070372_

Round 1

Reviewer 1 Report

There are two critical issues need to be improved.

1. The detailed information of the plants Gracilaria bursa-pastoris. First, is it a food or herb; second, the botany description and the related information are reader-attracted.

2. Title named Phytochemical Composition, but just the HPLC method was used to identify the chemical compounds. The NMR and MS are fundamental method to identify the chemical compounds, otherwise the Phytochemical Composition identification is not accurate.

3. The part of in vitro anti-diabetic effect is well by using biochemical parameters and molecular modeling.

English language fine. Minor editing of English language required.

Author Response

Revised version of Manuscript ID 2466704.                                     

Oujda, 14th of JUNE 2022

Dear Editor,

We have the pleasure of resubmitting the corrected version of our research article entitled "Study of the Phytochemical Composition, Antioxidant Proper-ties, and In vitro Anti-diabetic Efficacy of Gracilaria bursa-pastoris Extracts" by Safae Ouahabi, El Hassania Loukili, Nour Elhouda Daoudi, Mohamed Chebaibi, Mohamed Ramdani, Ilyesse Rahhou, Mohamed Bnouham, Marie-Laure Fauconnier, Belkheir Hammouti, Larbi Rhazi, Alicia Ayerdi Gotor, Flore Depeint and Mohammed Ramdani for publication in Journal of Marine Drugs.

Below is the itemized list of changes made in response to the reviewer's comments. We have thoroughly answered all of those comments and suggestions.

We hope that our paper's present version will meet your journal's requirements.

Yours sincerely,

OUAHABI Safae

Response to reviewer 1

We want to express our gratitude to reviewer 1 for his/her suggestions, which we feel have helped us improve our article's quality.

Reviewer 1: There are two critical issues need to be improved.

  1. The detailed information of the plants Gracilaria bursa-pastoris. First, is it a food or herb ; second, the botany description and the related information are reader-attracted.

R1- The authors agree with the reviewer. A paragraph containing detailed information on Gracilaria bursa-pastoris in the introduction has been added. (Underlined in yellow).

  1. Title named Phytochemical Composition, but just the HPLC method was used to identify the chemical compounds. The NMR and MS are fundamental method to identify the chemical compounds, otherwise the Phytochemical Composition identification is not accurate.

R1- Due to the unavailability of NMR and MS equipment in our department, it was difficult to analyze the compounds in Gracilaria bursa-pastoris extracts using these methods. Consequently, we relied on HPLC for phytochemical analysis, in particular HPLC-DAD, which offers better polyphenol selectivity.

  1. The part of in vitro anti-diabetic effect is well by using biochemical parameters and molecular modeling.

R1- We express our gratitude to reviewer 1.

Once again, we sincerely appreciate your time, effort, and constructive comments. We hope our revised manuscript successfully addresses all the reviewer 1 comments and meets the criteria for publication in Marine drugs.

Reviewer 2 Report

This manuscript has hight scientific quality. The topic of the research is in total agree with the scope of the journal. The experimental results support the conclusions of this work

Author Response

Revised version of Manuscript ID 2466704.                                     

Oujda, 14th of JUNE 2022

Dear Editor,

We have the pleasure of resubmitting the corrected version of our research article entitled "Study of the Phytochemical Composition, Antioxidant Proper-ties, and In vitro Anti-diabetic Efficacy of Gracilaria bursa-pastoris Extracts" by Safae Ouahabi, El Hassania Loukili, Nour Elhouda Daoudi, Mohamed Chebaibi, Mohamed Ramdani, Ilyesse Rahhou, Mohamed Bnouham, Marie-Laure Fauconnier, Belkheir Hammouti, Larbi Rhazi, Alicia Ayerdi Gotor, Flore Depeint and Mohammed Ramdani for publication in Journal of Marine Drugs.

Below is the itemized list of changes made in response to the reviewer's comments. We have thoroughly answered all of those comments and suggestions.

We hope that our paper's present version will meet your journal's requirements.

Yours sincerely,

OUAHABI Safae

Response to reviewer 2

Reviewer 2: This manuscript has hight scientific quality. The topic of the research is in total agree with the scope of the journal. The experimental results support the conclusions of this work.

We express our sincere gratitude to reviewer 2 for his/her positive feedback on the manuscript. We appreciate your assessment of its high scientific quality, and we are pleased to hear that the experimental results effectively support the conclusions drawn in the work. Your favorable evaluation is valuable to us, and we are sincerely grateful for your time and expertise in reviewing the manuscript.

Reviewer 3 Report

In this article, the authors evaluate the “Study of the Phytochemical Composition, Antioxidant Properties, and In vitro Anti-diabetic Efficacy of Gracilaria bursa-pastoris Extracts” In general, the work's novelty and quality is worthy to be published in marine drugs Journal.

-The Authors should write the above diabetes intro, global diabetes epidemiology, Common Side Effects of Diabetes Medication and the benefit of seaweeds in diabetes.

-The introduction part summarizes previous Seaweeds research, providing the wider context and background of the importance of the current study.

-Many pieces of the literature revealed that methanolic extracts have toxic and make some adverse effects. If so how does the author justifies the use of methanolic extracts?

-The statistical significance between the extract group was missing in Table 5 and Figure 3. Its significance statement may be explained in details. What ***?

-How about the total yield of Gracilaria bursa-pastoris Extracts

-IC50 should be written subscript and specie names in italics in this manuscript.

- Mention the concentration of Gracilaria bursa-pastoris and Ascorbic acid used in the antioxidant activity It is necessary to mention the concentration used in the study. In addition, discussed with previous literature “Antioxidant and Antibacterial Profiling of Pomegranate-pericarp Extract Functionalized-zinc Oxide Nanocomposite”. “Fermented Houttuynia cordata Against UVA and H2O2-Induced Oxidative Stress in Human Skin Keratinocytes”.

-Previous report about the anti-oxidative property of Gracilaria bursa-pastoris. I have seen some reports “Determination of the Phenolic Compounds and Antioxidative Capacity in Red Algae Gracilaria bursa-pastoris”. If so how do the authors claim the novelty of the study?

-Why do authors specifically target the polyphenols and flavonoids compounds? What about the antidiabetic role of other phytoconstituents?

 -Which phytochemical present in the Gracilaria bursa-pastoris Extracts are responsible for the activities used? Is it possible to predict?

-The reason should be stated for why the authors are keen to work under α-amylase and α-glucosidase in docking study?

Minor editing of English language required

Author Response

Revised version of Manuscript ID 2466704.                                     

Oujda, 14th of JUNE 2022

Dear Editor,

We have the pleasure of resubmitting the corrected version of our research article entitled "Study of the Phytochemical Composition, Antioxidant Proper-ties, and In vitro Anti-diabetic Efficacy of Gracilaria bursa-pastoris Extracts" by Safae Ouahabi, El Hassania Loukili, Nour Elhouda Daoudi, Mohamed Chebaibi, Mohamed Ramdani, Ilyesse Rahhou, Mohamed Bnouham, Marie-Laure Fauconnier, Belkheir Hammouti, Larbi Rhazi, Alicia Ayerdi Gotor, Flore Depeint and Mohammed Ramdani for publication in Journal of Marine Drugs.

Below is the itemized list of changes made in response to the reviewer's comments. We have thoroughly answered all of those comments and suggestions.

We hope that our paper's present version will meet your journal's requirements.

Yours sincerely,

OUAHABI Safae 

Response to reviewer 3 

We want to express our gratitude to reviewer 3 for his/her suggestions, which we feel have helped us improve our article's quality.

Reviewer 3: In this article, the authors evaluate the "Study of the Phytochemical Composition, Antioxidant Properties, and In vitro Anti-diabetic Efficacy of Gracilaria bursa-pastoris Extracts" In general, the work's novelty and quality is worthy to be published in marine drugs Journal.

  1. The Authors should write the above diabetes intro, global diabetes epidemiology, Common Side Effects of Diabetes Medication and the benefit of seaweeds in diabetes.

R3- A paragraph on the global epidemiology of diabetes, common side effects of diabetes medications and the benefits of algae for diabetes has been added in the introduction (underlined in green).

  1. The introduction part summarizes previous Seaweeds research, providing the wider context and background of the importance of the current study.

R3- Thank you for the time devoted in providing these insightful and vigilant comments.

  1. Many pieces of the literature revealed that methanolic extracts have toxic and make some adverse effects. If so how does the author justifies the use of methanolic extracts ?

R3- Despite the toxicity of methanolic extracts, the aim was to extract the maximum amount of polyphenols that are not soluble in other solvents and to compare them with the other extracts studied: aqueous and ethyl acetate.

  1. The statistical significance between the extract group was missing in Table 5 and Figure 3. Its significance statement may be explained in details. What ***?

R3- Statistical significance between extract groups has been added in Table 5 and Figure 3, and its significance statement has been explained in detail in the text (underlined in green).

  1. How about the total yield of Gracilaria bursa-pastoris Extracts

R3- The total yield of Gracilaria bursa-pastoris extracts is low and depends on several factors, including the extraction method, the solvent used, the plant material quality, the stress mode, and the extraction conditions.

  1. IC50 should be written subscript and specie names in italics in this manuscript.

R3- IC50 and specie names have been corrected in the manuscript.

  1. Mention the concentration of Gracilaria bursa-pastoris and Ascorbic acid used in the antioxidant activity It is necessary to mention the concentration used in the study. In addition, discussed with previous literature "Antioxidant and Antibacterial Profiling of Pomegranate-pericarp Extract Functionalized-zinc Oxide Nanocomposite". “Fermented Houttuynia cordata Against UVA and H2O2-Induced Oxidative Stress in Human Skin Keratinocytes”.

R3- The concentration of Gracilaria bursa-pastoris and ascorbic acid used for antioxidant activity and Discussion with the literature have been added in the manuscript (underlined in yellow).

  1. Previous report about the anti-oxidative property of Gracilaria bursa-pastoris. I have seen some reports "Determination of the Phenolic Compounds and Antioxidative Capacity in Red Algae Gracilaria bursa-pastoris". If so how do the authors claim the novelty of the study ?

R3- In the article entitled "Determination of the Phenolic Compounds and Antioxidative Capacity in Red Algae Gracilaria bursa-pastoris," the authors only studied the antioxidant activity of aqueous, hexanic and ethanolic extracts by TAC (Total Antioxidant Capacity). In contrast, our study evaluated the antioxidant activity of DPPH and β-carotene bleaching assay on other extracts such as ethyl acetate and methanol. It is important to emphasize that antioxidant activity is a complex procedure that generally occurs via several mechanisms and is influenced by many factors, which cannot be fully described by a single method. It is, therefore, essential to carry out more than one type of antioxidant capacity measurement to consider the different mechanisms. Thus, in our study, the antioxidant potential of G. bursa-pastoris extracts was assessed and confirmed using two methods with two different mechanisms: the 2,2-diphenyl-1-picrylhydrazyl (DPPH) radical scavenging assay and the β-carotene bleaching method.

  1. Why do authors specifically target the polyphenols and flavonoids compounds ? What about the anti-diabetic role of other phytoconstituents ?

R3- The potential of Polyphenols and flavonoids to promote health and prevent or treat certain diseases makes them interesting targets for authors interested in nutrition, medicine, and wellness. Numerous studies have indicated that these compounds possess a variety of health benefits, including antioxidant, anti-diabetic, anti-cancer, and cardiovascular protective properties.  We have also expanded the discussion by adding sections that describe information about the antidiabetic effect of other phytoconstituents present in our extracts, such as quercetin, kaempferol... We have added the response to the comment in the discussion section and we have highlighted the responses in green.

  1. Which phytochemical present in the Gracilaria bursa-pastoris Extracts are responsible for the activities used ? Is it possible to predict ?

R3- The phytochemicals responsible for the alpha-amylase and glucosidase inhibitory effects in Gracilaria bursa-pastoris extracts have not been studied extensively from a scientific perspective. Furthermore, we would like to provide some general information about phytochemicals commonly found in various plants, including those present in Gracilaria bursa-pastoris extracts, which have been reported to exhibit inhibitory activities against alpha-amylase and alpha-glucosidase enzymes. We have added the response to the comment in the discussion section and we have highlighted the responses in green.

  1. The reason should be stated for why the authors are keen to work under α-amylase and α-glucosidase in docking study ?

R3- Thank you, dear reviewer, for your question. α-amylase and α-glucosidase play an important role in the regulation of blood sugar and are closely related to the antidiabetic effect. To investigate these roles further, we conducted an in vitro study specifically focusing on the activities of α-amylase and α-glucosidase. Following our in vitro study, we proceeded with an in-silico study, which complemented and validated our findings from the in vitro experiments.

Once again, we sincerely appreciate your time, effort, and constructive comments. We hope our revised manuscript successfully addresses all the reviewer 3 comments and meets the criteria for publication in Marine drugs.